# A Novel Isolate of *Bacillus cereus* Promotes Growth in Tomato and Inhibits *Clavibacter michiganensis* Infection under Greenhouse Conditions

**DOI:** 10.3390/plants10030506

**Published:** 2021-03-09

**Authors:** Nallely Solano-Alvarez, Juan Antonio Valencia-Hernández, Enrique Rico-García, Irineo Torres-Pacheco, Rosalía Virginia Ocampo-Velázquez, Eleazar Máximo Escamilla-Silva, Ana Luz Romero-García, Ángel G. Alpuche-Solís, Ramón Gerardo Guevara-González

**Affiliations:** 1C.A Ingeniería de Biosistemas, Facultad de Ingeniería, Campus Amazcala, Universidad Autónoma de Querétaro, Carretera Chichimequillas s/n km1, El Marques, Querétaro 76265, Mexico; nallely.solanoa@gmail.com (N.S.-A.); qaantoniovalencia@hotmail.com (J.A.V.-H.); ricog@uaq.mx (E.R.-G.); irineo.torres@uaq.mx (I.T.-P.); rosov05@yahoo.com.mx (R.V.O.-V.); 2Departamento de Ingeniería Quimica, Tecnologico Nacional de Mexico, Ave. Tecnologico y A. Garcia-Cubas, S/N, Celaya, Guanajuato 38010, Mexico; chiquisdrilis67@gmail.com; 3División de Biología Molecular, Instituto Potosino de Investigación Científica y Tecnológica, San Luis Potosi 78216, Mexico; rgggon@hotmail.com (A.L.R.-G.); alpuche@ipicyt.edu.mx (Á.G.A.-S.)

**Keywords:** PGPB, bacterial canker, crop protection

## Abstract

The need to produce food in a sustainable way to counteract the effects of excessive use of agrochemicals opens the door to the generation of new technologies that are not based on fossil fuels and are less toxic to ecosystems. Plant growth-promoting bacteria (PGPB) could represent an alternative to chemical biofertilizers and pesticides offering protection for biotic and abiotic stresses. In this work, a bacterial isolate from roots of castor bean (*Ricinus communis*) was identified and named as *Bacillus cereus* strain “Amazcala” (B.c-A). This isolate displayed the ability to solubilize inorganic phosphate and produce gibberellic acid (GA3). Moreover, this bacterium provided significant increases in height, stem width, dry weight, and total chlorophyll content in tomato plants. Interestingly, B.c-A also significantly decreased the severity of bacterial canker disease on tomato caused by *Clavibacter michiganensis* (*Cmm*) in preventive disease assays under greenhouse conditions. Based on our results, B.c-A can be considered as PGPB and a useful tool in *Cmm* disease control on tomato plant under greenhouse conditions.

## 1. Introduction

The rapid growth of the world population, as well as the need to produce food in a more sustainable way without the use of synthetic agrochemicals, has brought, in recent years, a search for alternatives substances [1]. One of these alternatives could be the use of beneficial microorganisms like plant growth-promoting bacteria (PGPB) that enhance plant growth involving direct mechanisms that improve nutrient uptake, regulation of phytohormones [2], and ability to cope with both biotic and abiotic stresses to which the plants are exposed [3].

PGPB’s have the ability to directly stimulate the development of plants by means of several mechanisms such as nitrogen supply through the process of biological fixation of atmospheric nitrogen, production of growth-regulating substances, solubilization of minerals, induction of systemic resistance to pathogens, and inhibition of the growth of pathogenic organisms [1,4].

The term PGPB includes three types of soil bacteria depending on their way of life: free-living bacteria that inhabit the area around the root (rhizosphere), those that colonize the root surface (rhizoplane), and endophytic bacteria that colonize inside the root [5]. This classification is not exclusive, since any individual bacterial strain can adopt any of the three lifestyles depending on the conditions of the soil environment and the root of the host involved [6].

There is evidence that many genera of bacteria can be a PGPB, for example *Bacillus amyloliquefaciens*, which confers protection in rice cultivation to stress caused by salinity and drought [7], *Kosakonia radicincitans*, which affects the number of amino acids, sugar and volatile compounds improving the flavor of the tomato [8], or even *Variovorax paradoxus*, which helps reduce fertilization in drought conditions of this same vegetable [9]. There also exist reports that the growth of tomato plants is enhanced by the inoculation of *B. pumilus* under additional N supply [10].

Tomato (*Solanum lycopersicum*) is one of the most cultivated vegetables worldwide over the last century [11]. In Mexico, tomato production in 2016 was more than 2.769 million tons, which makes it the main export crop growing at an annual rate of 6.5%. The countries where tomato is mainly exported from Mexico are the United States, Canada, and Japan with a value of one billion USD, countries that have increasingly stricter regulations regarding the residuality of agrochemicals that are used in their production [11]. Bacterial canker of tomato caused by *Clavibacter michiganensis* ssp. *michiganensis* (*Cmm*) is considered the most important bacterial disease affecting tomato crops worldwide [12]. This pathogen can enter the host plant through wounds and natural openings in leaves, stems and roots, and broken stomata [13].

In addition, *Cmm* can infect seeds externally, or it can enter seed coat and endosperm, and invade the xylem vessels, followed by a systemic infection of the host [13], when *Cmm* invades vascular tissues, initially causing wilting of leaves and leaflets, followed by wilting of the whole plant, necrosis, and cankers on the stems and petioles [14]. This infectious disease can spread rapidly and result in a severe loss of the productivity [15].

As for other bacterial diseases, currently there are not efficient chemical control strategies to cope with *Cmm* diseases [12]. Moreover, there are no tomato cultivars resistant to *Cmm* disease and breeding genetic progress for resistant lines is modest [16]. Hence, there is a need to generate technology according to the new demands of the market faced by producers, generating products based on PGPB’s isolates where mainly these could confer some protection before the presence of the disease [17].

The present work aimed to evaluate the plant growth promotion and protecting effect against bacterial canker disease caused by *Cmm*, of a novel isolate of *Bacillus cereus* strain Amazcala (B.c-A) isolated from roots of castor bean (*Ricinus communis*) on tomato plants under greenhouse conditions. Interestingly, B.c-A displayed biostimulant properties on tomato plants as well as protection against bacterial canker in vitro and in vivo under greenhouse conditions.

## 2. Results

### 2.1. Isolation of Bacteria

A soil sample was taken from the rhizosphere region of a castor bean plant, having reports of surviving different types of environmental stresses during at least 3 years in campus Amazcala of Autonomous University of Querétaro (Central México). From these samples, 5 types of colony morphologies were obtained, only one of them (named as isolate RR2) was selected for further studies in this work based on the higher appearance frequency in comparison to the other colonies. The morphology of RR2 colonies in Luria Bertani (LB) plates consisted in a circular shape and smooth appearance, irregular edge and creamy consistency, gram-positive bacilli, and identified as *Bacillus cereus* according to 16S ribosomal gene DNA sequencing (96% identity to *B. cereus* P0021 in BLASTn analysis). From herein, this bacterial isolate will be called B.c-A.

### 2.2. Inorganic Phosphate Solubilization Tests

B.c-A, displayed a transparent halo around the colonies grown in Pikovskaya medium in comparison with the control bacteria (a commercial product based on *B. subtilis*), thus indicating that this isolate is a phosphate solubilizer (Figure 1). This latter difference between the halos in both treatments resulted statistically significant according to Tukey test (*p* = 0.05).

### 2.3. Plant Growth Promotion in Tomato Plants

After one month of weekly inoculations on tomato plants, no significant differences between the treatments with the two different concentrations of B.c-A were observed (Figure 2). However, both bacterial treatments were significantly different with respect to control without bacteria. Both treatments increased plant height, dry weight, stem width, and chlorophyll content (Figure 2). Thus, B.c-A showed significant growth promotion of tomato plants in both concentrations evaluated in the study. The fact that plant height was promoted by B.c-A in the tomato plants suggested the possibility that this microorganism produce gibberellins. Thus, a gibberellic acid (GA3) measurement test was carried out in B.c-A, showing that this bacterium was able to produce high levels of GA3 (0.051 ng·mL^−1^, data not shown), likely suggesting that the phenotype of higher tomato plants was caused by the production of gibberellins.

### 2.4. Enzymatic Plant Stress-Response

B.c-A applications onto tomato plants displayed a significant increase in enzyme activities related to stress response and defense (Figure 3). The inoculation of the bacteria significantly increased enzyme activities related to ROS scavenging (Superoxide Dismutase, SOD and Catalase, CAT), as well as phenylpropanoids biosynthesis (Phenylalanine ammonia-lyase, PAL) (Figure 3). The increase in these enzyme activities suggested that B.c-A induced the innate immune system on tomato plants. The latter asseveration also suggested that it is likely that these treated plants might display increased tolerance to biotic stresses, based on the fact that the induction of these stress-response enzymatic systems is associated with increase in plant tolerance to diseases caused by several pathogens [17].

### 2.5. Gene Expression-Associated with Plant Defense

In order to study, at a molecular level, the stress-response induced by B.c-A on tomato plants mentioned above, the expression of 2 genes related to plant defense was carried out. The chalcone synthase gene (*chs*) involved in phenylpropanoids biosynthesis (flavonoids) and the salicylic acid defense pathway gene marker *pr1a* were analyzed. On the one hand, the expression of *chs* was significantly induced by B.c-A (Figure 4). On the other hand, the relative gene expression of *pr1a* significantly decreased when tomato plants were treated with B.c-A in comparison with control plants (Figure 4).

### 2.6. Antagonistic Activity of B.c-A against Cmm In Vitro

The in vitro antagonistic activity of B.c-A against three different isolates of *Cmm* (sp, AcR42 and 1569) was evaluated (Figure 5). B.c-A displayed significant antagonistic activity against the 3 strains of *Cmm* evaluated (Figure 5). The supernatant of B.c-A grown in LB and MM evaluated in these tests displayed no significant differences in the antagonistic activity against the 3 strains of *Cmm* (Figure 5). In order to evaluate possible antagonistic effects of B.c-A on other types of bacteria, *E. coli* (DH5α) and an isolate of *Pseudomonas aeruginosa* were also included in these assays, showing no affection by B.c-A (data not shown).

### 2.7. Activity of B.c-A against Tomato Bacterial Canker in Greenhouse Tests

In order to evaluate the efficacy of B.c-A in protecting tomato plants against bacterial canker disease, two independent experiments were carried out under greenhouse conditions. For these studies *Cmm* AcR42, which is the most pathogenic isolate of the three evaluated in this work was used. The results displayed that plant size in control plants treated only with B.c-A was significantly the highest in comparison with the rest of the treatments (Figure 6, Panel A). This figure also shows that the treatment with B.c-A + *Cmm* AcR42 displayed significant higher plants than treatments only with *Cmm* AcR42 and those with *Cmm* + chemical product (Figure 6, Panel A). The severity of disease significantly decreased in approximately 50 % with the application of B.c-A, and 25 % with the chemical product in comparison control disease treatment (Figure 6, Panel B). Moreover, B.c-A even displayed significant decrease in severity level of disease in comparison with the chemical control (Figure 6, Panel B). The morphological aspect of B.c-A treated plants at 45 days post-germination displayed the beneficial effects of the bacterial treatments (Figure 7).

## 3. Discussion

The deficiency of phosphorus (P) in the soil influences the delay in maturity in the development of the plants, decreasing the yield of the harvest [18]. The Pikovskaya medium contains tricalcium phosphate, which is an inorganic poor soluble source of P and is not assimilable by the plant. However, a transparent halo formed around a bacterial colony indicates capacity of dissolving this inorganic salt, thus becoming an assimilable form. As shown, B.c-A, was able to produce halo in Pikovskaya medium, indicating that this isolate possesses the capacity to solubilize P from unavailable sources. This activity has been reported in several *Bacillus* species, including *B. cereus*, mediated by producing several organic acids as lactic, acetic, and gluconic acids [19]. Future studies with P-solubilizing activity of B.c-A will consider the use of harder-to-dissolve compounds, such as Fe-P or Al-P in order to establish, in a more robust way, the phosphorus solubilization capacity of B.c-A [20].

Several authors have reported that *Bacillus* species (including *B. cereus*) are active producers of several gibberellins, significantly increasing plant height in pepper plants [21]. B.c-A displayed plant growth promotion activity on the variety of tomato evaluated in this study, suggesting the possibility that this isolate is a producer of gibberellins that in fact were tested in the present work. As mentioned in the results section, this isolate produced high amounts of GA3 in culture medium. This gibberellic acid production activity might also be found when testing these bacteria in different plant species from the one they were isolated (as is the case of B.c-A that was isolated from castor bean but improved performance in tomato). The PGPB activity of several microorganisms on different crops with a specific bacterial genus has already been reported in grasses, *fabaceae,* and some weeds [2,22,23].

Earlier literature suggested that in most of the cases, PGPBs trigger defense-responses, leading to the production of biochemical and molecular responses such as increase in enzyme activities and gene expression associated with oxidative stress response and phenylpropanoids biosynthesis [17]. In this study the presence of B.c-A increased PAL activity and *chs* gene expression, both associated with phenylpropanoids biosynthesis. Moreover, antioxidant enzymes as SOD and CAT activities were also significantly increased by B.c-A applications in tomato plants. Thus, it is clear that this bacterium induced defense-response in the treated tomato plants evaluated in this work.

In vitro antagonistic activity against *Cmm* was shown by B.c-A grown on LB or MM, indicating that both media evaluated in this study did not affect the physiology of this bacterium regarding the antagonistic activity against *Cmm*. This type of knowledge might be important for future industrial application purposes of B.c-A, because it is important to search for cheaper culture mediums, in order to diminish costs of large-scale production of PGPBs. Moreover, it has been reported that strains of *B. subtilis* and *Pseudomonas* sp display antagonistic activity in vitro and *in vivo*, respectively, against *Cmm* and the bacterial canker in tomato [24] and [25]. To our knowledge, the results of the present work are the first report showing that a strain of *Bacillus cereus* (as B.c-A) displays in vitro and in vivo antagonistic activity against *Cmm* and the canker disease provoked by this pathogen in tomato.

Finally, it has been shown that several PGPB´s like *Bacillus* sp can confer disease resistance to plants through induced systemic resistance (ISR) regulated by jasmonic acid, as reported by Dejellout [26]. Actually, it has been shown that *B. subtilis* SL18r was able to increase the resistance of tomato plants against the invasion of *B. cinerea* in such an effective manner that did not require for activating a series of pathogenesis-related proteins and best fitted plant growth [27].

Moreover, it has also been reported that *Bacillus* sp as PGPB, induces systemic acquired resistance (SAR) pathway, that normally is induced by infection of virulent, avirulent, or nonpathogenic microorganisms and it is regulated by salicylic acid (SA) [28]. In the present study, B.c-A showed significant induction of biochemical and molecular markers related to induced or systemic resistance (ISR or SAR). Distinguishing ISR from SAR with a PGPB (as B.c-A) as an inductor is not obvious. Trying to analyze the pathway by which B.c-A induced the observed resistance phenotype to bacterial canker in tomato plants in the present work, the gene expression analysis of a pathogenesis-related gene as *pr-1a*, considered as a SAR marker, displayed a repression by the B.c-A treatment (Figure 5). The latter result strongly suggested that B.c-A induced bacterial canker resistance in this study by a SA-independent pathway. In addition, B.c-A significantly induced *chs* gene, related to flavonoids biosynthesis. Flavonoids have been related to crop protection in several plant-microbe interaction models [3]. Thus, it is likely that at least partially, the tolerance to bacterial canker in tomato induced by B.c-A is related to phenylpropanoids, especially flavonoids production.

Based on the above-mentioned results, B.c-A might be an interesting PGPB alternative to be used as an element in the sustainable production of tomato under greenhouse conditions at least in Central Mexico.

## 4. Materials and Methods

### 4.1. Media for Bacterial Cultivation

For bacterial cultivation during the study, Luria Bertani (LB) and M-9 minimal medium (MM) was used. Only in the case of inorganic phosphate solubilization assays the Pikovskaya medium was used. Concentration of bacteria was determined in all cases by serial dilutions and expressed as CFU mL^−1^.

### 4.2. Isolation of PGPB

Root samples were collected from a castor bean plant (*Ricinus communis*) located at the facilities of the Autonomous University of Querétaro Amazcala campus Amazcala (20°09′ N, 46 100°40′ W to 1900 m) where the predominant climate is semi-desertic with average temperatures of 18 °C and average rainfall of 200 mm. Castor bean was selected for searching PGPB’s based on historical observations in this plant species for biotic and abiotic tolerance for several years in campus Amazcala. One g of root was immersed in 9 mL of sterile saline solution and processed further for isolation of bacteria by serial dilutions up to dilution factor of 10^−6^ were prepared followed by spreading on for 2–3 days [2]. The isolated colonies were subcultured individually in LB agar and cryopreserved at −80 °C in 40% glycerol for future studies. Five types of colonies were obtained and named as: RR1, RR2, RR3, RR4, and RR5. The RR2 isolate was the only one evaluated in the present work, based on that was observed with the highest number of colonies in the different dilutions.

### 4.3. PCR Amplification of 16S rRNA and Sequence Analysis

DNA purification from the RR2 isolate was carried out by a modification of the Gilbert protocol by replacing the Tris buffer with CTAB [28]. The genomic DNA obtained was stored at −20 °C for analysis of the 16S ribosomal gene sequence. For the amplification of the 16S gene fragment by PCR, primers 27 F (5′-GAGTTTGATCCTGGCTCAG-3′) and 1492 R (5′-GGTTACCTTACGACTT-3′) were used [29]. The reaction was carried out as follows: initial denaturation at 95 °C for 10 min, 30 cycles at 95 °C for 30 s, 50 °C for 30 s, and 72 °C for 45 s and final extension 72 °C for 5 min. The products were separated by electrophoresis in 1.2% agarose gel and observed one band at 1500 bp. The 16S DNA sequence obtained from the amplicon was compared with others in the Gen Bank database using NCBI BLAST program in http://www.ncbi.n1m.nih.gov/blast/Blast.cgi (accessed on 6 February 2021). This sequence was submitted to the GenBank/EMBL/DDBJ database under the accession number MK941165.

### 4.4. Detection of GA3 by HPLC

The determination of gibberellins was made by HPLC [30]. Samples of 20 mL of supernatant of Luria Bertani broth in which RR2 isolate was growing for 24 h was collected and treated as described [30]. In these studies, uninoculated LB was used as experimental control. A solution of GA3 (Sigma-Aldrich) was used as a standard for these analyses. Quantification was performed on an Agilent 1200 series HPLC, UV-VIS 6120 Quadrupole using an Eclipse plus C18 5 μm particle size column. All analyses were carried out in triplicate.

### 4.5. Inorganic Phosphate Solubilization

The ability to solubilize inorganic phosphate was tested by depositing 5 μL of the overnight growth of the isolate in Luria Bertani (LB) agar and incubation at 28 °C ± 2 °C on the center of petri dishes with Pikovskaya agar (PVK) for 10 days at 30 °C ± 2 °C [31]. A commercial strain of *Bacillus subtilis* (PROCOBI, Queretaro, México) was evaluated as positive control. The formation of transparent zones around the bacterial colonies was an indication of the solubilization of the inorganic phosphate contained in the PVK agar. This analysis was carried out in two independent experiments in triplicate and the solubilization was determined by measuring the diameter of the transparent zones of the solubilization halo generated in the PVK medium using a vernier [32]. The study was carried out in two independent experiments (*n* = 3, total in two experiments *n* = 6).

### 4.6. Effect on Plant Growth Promotion in Tomato

The effect of B.c-A on the growth of tomato seedlings cv. “Ailsa Craig” was evaluated in a greenhouse of 150 m^2^. The average temperature of the greenhouse was 29 °C ± 3 °C and the relative humidity 70%, the photoperiod corresponded to natural light of 12:12 light-dark cycles. Plants were irrigated daily to field capacity (100 mL per day) and the nutrient solution used was Steiner [33].

B.c-A was grown in LB broth with constant agitation at 180 rpm at 30 °C for 48 h. At the end of the logarithmic phase, the bacterial culture was centrifuged at 300 rpm for 3 min and the supernatant was discarded, the pellet was resuspended in sterile water. The aqueous suspension was diluted until it reached a concentration of 1.5 × 10^8^ CFU mL^−1^ determined by serial dilutions. The study was carried out in two independent experiments with a randomized block design with three replications and experimental units of six-plants. The treatments consisted in the immersion of seeds during 5 min in sterile water (negative control) and two treatments with immersion during the same time in 2 different concentrations of B.c-A (1 × 10^4^ CFU mL^−1^ and 1 × 10^8^ CFU mL^−1^). Then, 0.5 mL of the bacterial suspension per kg of substrate was inoculated at the base of the plant (a 3:1 mix of peat moss:vermiculite) where the tomato plants were growing. The first inoculation in the substrate was carried out when the plants generated their first true leaves (day 3) and the subsequent inoculations were weekly for 45 days post-germination. Basal stem width and plant height were measured using a vernier, as well as dry weight (differential weight of the 6 whole plants for each treatment after drying in oven at 50 °C during 24 h), and amount of chlorophyll (measured as SPAD units, Minolta, Osaka, Japan) were determined.

### 4.7. Antagonism Test against Clavibacter Michiganensis In Vitro

Three different strains of *Cmm* were isolated in LB plates from diseased tomato plants in production areas affected by these bacteria in Central Mexico (Acr42, 1565 and sp) and provided by molecular biology department at IPICYT. Five hundred µL of a suspension of each *Cmm* isolate grown in liquid LB (1 × 10^8^ CFU mL^−1^), were dispersed on the surface of LB agar plates. After 5 min, filter paper discs of 5 mm in diameter previously immersed in a suspension of B.c-A (1 × 10^8^ CFU mL^−1^) grown either on LB or MM (to evaluate the effect of bacteria grown in minimal nutrient medium) [34] were placed on the plates previously inoculated with *Cmm* [35]. The plates were incubated at 28 °C ± 2 °C for 48–72 h or until a zone of inhibition was observed. LB was used as a negative control, and the commercial product Qbacter^®^ (Quimcasa, Querétaro, México) as a positive control. In order to valuate possible general inhibitory effects by other bacteria, an *E. coli* strain (DH5α) and an isolate from Central México of *Pseudomonas aeruginosa* were also included in the study. The inhibitory effect was determined as a straight line measured from the border of the filter paper until the border of the inhibition zone in the plates. Three independent experiments were carried out for these assays with three replicates.

### 4.8. Evaluation of Antagonistic Activity of B.c-A against Clavibacter Michiganensis under Greenhouse Conditions

The antagonistic effects under greenhouse were carried out in two independent experiments. The surface of the greenhouse was 15 m^2^, with average temperature of 29 °C ± 3 °C and relative humidity 80% and a natural light photoperiod of 12 h:12 h (light-dark). Thirty-one days post-germination tomato plants were inoculated with B.c-A by both spraying onto leaves and applying in the substrate 1 mL per kg of substrate of an inoculum grown in MM (1 × 10^8^ CFU mL^−1^) in three weekly applications. Two days later, the first B.c-A application, the most pathogenic *Cmm* strain (AcR42) was inoculated by puncturing the stem with a bacterial suspension grown in liquid LB (1 × 10^8^ CFU mL^−1^). As chemical control, Terra-Cu^®^ product (CYR-agroquimica company, Morelos, México) was applied according to manufacturer instructions (50 mL per plant in the substrate at a concentration of 140 ppm). Meanwhile, another chemical control was the product Oxymeth^®^ (cuprous oxide, Cuprosa, Jalisco, Mexico) which was sprayed (3.8 mL) to the leaves in a concentration of 0.03 g mL^−1^ when the first symptoms of disease appeared. As a negative control, plants with no pathogen inoculation were used. The experimental set up of the two independent experiments consisted in a completely randomized design, using 9 plants (sowed in pots of 450 cm^3^ using peat moss as substrate, Hydro Environment, México City, México) as experimental unit. The variables measured were plant height (with a rule) and the severity of bacterial canker disease using the Soylu scale at 30 and 45 days post-germination [36].

### 4.9. Stress Response-Antioxidant Enzyme CAT, PAL, SOD Activities Measurements

Total soluble proteins (TSP) were extracted from lyophilized leaves (50 mg) ground with 1.5 mL extraction buffer containing 0.05 M phosphate buffer (pH 7.8), 0.1 mM Na2EDTA. The samples were centrifuged at 12,000 rpm for 15 min at 4 °C. Protein concentration was determined using the Bradford Reagent (Merck, Darmstadt, Germany) and bovine serum albumin (Merck, Darmstadt, Germany) as standard. The supernatant obtained for the TSP assay was used to quantify catalase (CAT), phenylalanine ammonia lyase (PAL) and superoxide dismutase (SOD) activities. CAT activity was assayed by following the initial rate of H_2_O_2_ degradation for 120 s monitored at 240 nm, Enzyme activity was determined [37] with some modifications. PAL was determined by spectrophotometry at 290 nm quantifying the cinnamic acid formed from the catalysis of L-phenylalanine [38]. SOD activity was determined spectrophotometry at 560 nm by measuring the photochemical reduction of nitroblue tetrazolium (NBT) [39].

### 4.10. Gene Expression-Associated with Plant Defense

Molecular markers for phenylpropanoids biosynthesis (chalcone synthase, chs, accession number NM_001247104.2) and a molecular marker of salicylic acid plant defense pathway (pathogenesis-related protein 1a, pr-1a, accession number XM_004242627.4) were evaluated in the study as indicators of immunity induction in the tomato plants. The actin gene (act, accession number AB199316.1) was used as housekeeping control. Triplicate upper leaf samples were ground in liquid nitrogen, and RNA extracted (PureZOL™ RNA Isolation, Bio-Rad). The integrity of the extracted RNA was checked on agarose gel electrophoresis, and its purity and concentration were assessed by a NanoDrop. One μg of total RNA of high purity (260/280 nm absorbance ratio above 2.0 and 260/230 nm absorbance ratio 1.8–2.0) was used to synthesize cDNA following the manufacturer instructions (Thermo Fisher™). The cDNA was diluted to 400 ng uL^−1^ and stored in −20 °C for qPCR until further analysis. Primer sequences used to amplify pr-1a (forward 5′-GCCAAGCTATAACTACGCTACCAAC-3′; reverse: 5′- GCAAGAAATGAACCACCATCC-3′), chs (forward 5′-CCAAGGACTTGGCTGAGAAC-3′; reverse 5′-TATCGGGGACAAGAGTTTGG-3′) and actine (act) as housekeeping control gene (forward 5′-ATGTGACGTGGATATTAGGAAAG-3′; reverse: 5′-AGGGAAGCCAAGATAGAGCC-3′) were used. The qPCR analysis was carried out in a real Time System (BIORAD Laboratories) using sybrgreen as fluorophore. Reaction conditions for all the genes were: 5 s at 94 °C and 40 cycles of 5 s at 94 °C and 30 s at 57 °C. The relative gene expression levels were calculated using the ΔΔCt method [40].

### 4.11. Statistical Analysis

For all the experiments, a normality test of data (Anderson-Darling test, *p* = 0.05) was carried out. If data were not in the normality, non-parametric tests as Kruskal–Wallis H test (*p* = 0.05) was carried out instead of one-way ANOVA (*p* = 0.05). All the results obtained were expressed by the means plus/minus their standard errors. The differences between treatments were determined by one-way ANOVA, and differences between the means were determined by Tukey test (*p* = 0.05) using the GraphPad PRISM version 7 software. Only in antagonistic tests in greenhouse for *Cmm* severity data (as ordinal variable and out of normality), the statistical analysis was carried out using the Kruskal–Wallis H test (XLSTAT-Excel) and the Dunn’s tests (*p* = 0.05) to determine differences between treatments.

### 4.12. Ethical Statement

The authors declare that they have no conflict of interest. This manuscript does not contain any studies with human or animal participants performed by any of the authors, and this project was approved by ethics committee from School of Engineering from Autonomous University of Querétaro (Queretaro, Mexico) with approval reference number CEAIFI-100-2018-TP.

## Figures and Tables

**Figure 1 plants-10-00506-f001:**
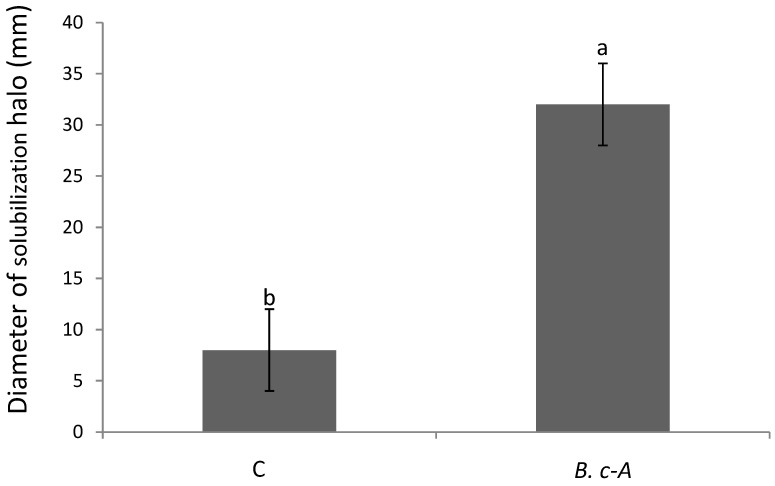
Phosphates solubilization activity of *B. cereus* “Amazcala”. As positive control of phosphate solubilization (Control, C), a *Bacillus subtilis* commercial strain with this property was used (see methods). Different letters in each bar indicates significant difference according to Tukey test (*p* = 0.05). The results are the average of two independent experiments (*n* = 3, total in two experiments *n* = 6).

**Figure 2 plants-10-00506-f002:**
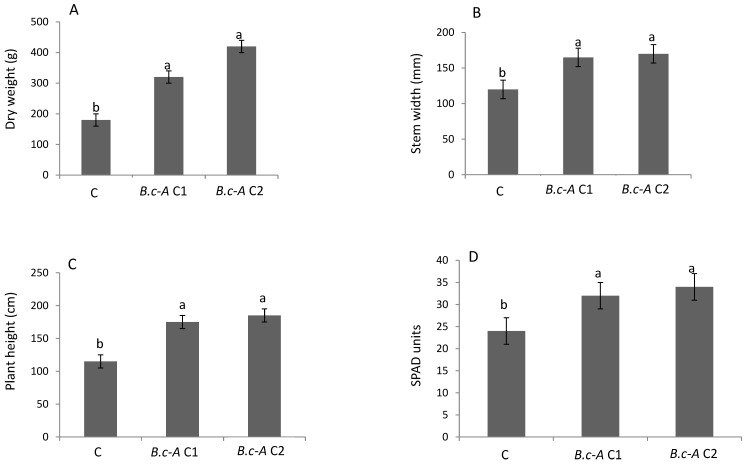
Effect of two colony-forming units (CFU) levels of *B. cereus* “Amazcala” on tomato performance. Panel **A**, Dry weight; Panel **B**, Stem width; Panel **C**, Plant height; Panel **D**, Chlorophyll levels as SPAD units. Simbology: C, control plants no-treated with *B. cereus* Amazcala; B.c-A, *B. cereus* Amazcala tested at 1 × 10^4^ CFU mL^−1^ (C1) or at 1 × 10^8^ CFU mL^−1^ (C2), respectively. Different letters in each bar for each panel indicates significant difference according to Tukey test (*p* = 0.05). The results are the average of two independent experiments (*n* = 6 for each experiment, total in two experiments *n* = 12).

**Figure 3 plants-10-00506-f003:**
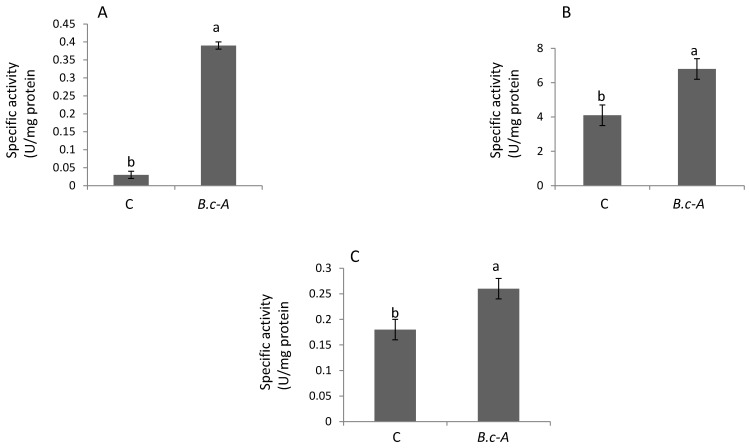
Stress-response enzyme activities in tomato plants treated with *B. cereus* “Amazcala”. Panel **A**, superoxide dismutase (SOD); Panel **B**, catalase (CAT); Panel **C**, phenylalanine ammonia lyase (PAL). Simbology: C, Control plants mean plant non-treated with the bacterium; B.c-A, plants treated with B. cereus Amazcala. Different letters in each bar for each enzyme activity indicates significant difference according to Tukey test (*p* = 0.5). The results are the average of 2 independent experiments (*n* = 3, total in two experiments *n* = 6).

**Figure 4 plants-10-00506-f004:**
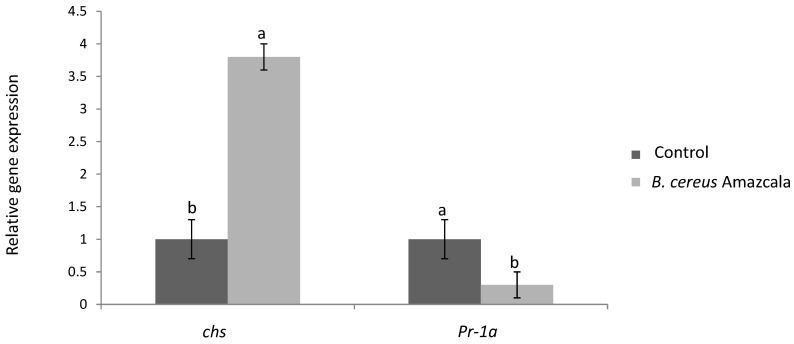
Relative gene expression of chalcone synthase (*chs*) and pathogenesis-related protein-1a (*pr-1a*). Different letters in each bar indicates significant difference according to Tukey test (*p* = 0.05). The results are the average of two independent experiments (*n* = 3, in total *n* = 6). Control plants mean plant non-treated with the bacterium.

**Figure 5 plants-10-00506-f005:**
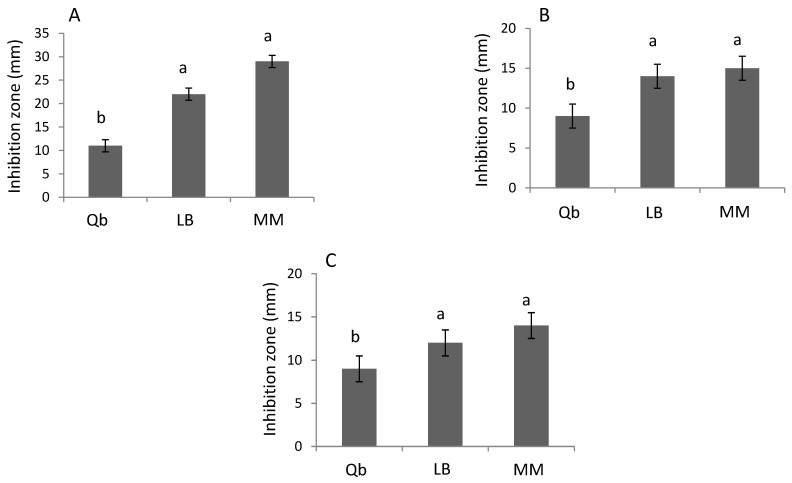
Size of the inhibition zone in the in vitro antagonistic tests of *B. cereus* Amazcala against 3 isolates of *Cmm*. Panel **A**, against isolate sp; Panel **B**, against isolate AcR42; Panel **C**, against isolate 1565. *B.c-A* was grown on Luria Bertani (LB) or minimal M9 medium (MM). As positive control against *Cmm*, a commercial product (Qbacter, Qb) was evaluated. Different letters in each bar for each panel indicates significant difference according to Tukey test (*p* = 0.05). The results are the average of three independent experiments (*n* = 3 for each experiment, total in three experiments *n* = 9).

**Figure 6 plants-10-00506-f006:**
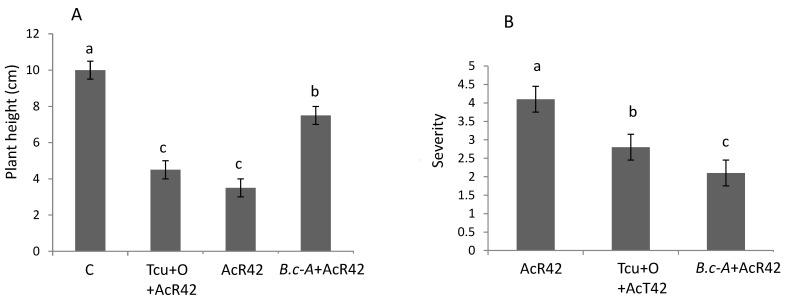
Plant height (Panel **A**) and severity (Panel **B**) of bacterial canker disease at 30 days post-inoculation in tomato plants treated with *Cmm* and/or *B. cereus* Amazcala under greenhouse conditions. As positive chemical control of bacterial canker, a commercial product (Terra-Cu-Oxymet) was evaluated. Control in panel A refers to plants inoculated only with *B. cereus* Amazcala. Simbology: C, Control; Tcu+O, Terra-Cu-Oxymet + *Cmm* AcR42; AcR42, *Cmm* isolate AcR42; Bc-A, *B.cereus*-Amazcala. Different letters in each bar indicate significant difference according to Tukey test in panel A, and Kruskal–Wallis H test in panel B, using in both tests a significance of *p* = 0.05. The results are the average of 2 independent experiments (*n* = 9 for each experiment, total in two experiments *n* = 18).

**Figure 7 plants-10-00506-f007:**
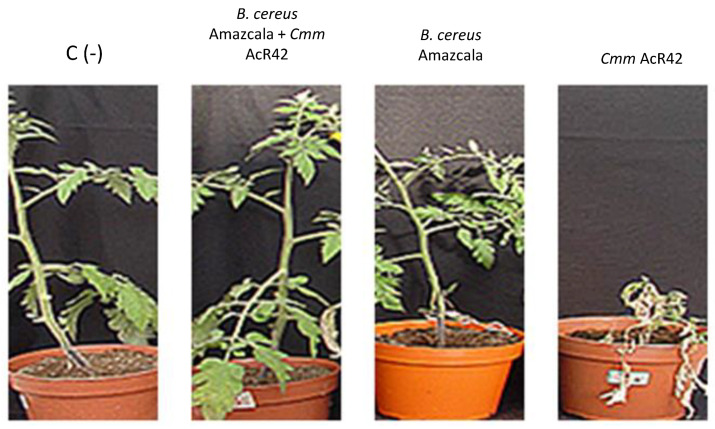
Typical symptoms of the tomato plants preventively treated with *B. cereus* Amazcala and infected with *Cmm* AcR42. C (-), negative control treated neither with *B. cereus* Amazcala nor *Cmm* Ac42; *Cmm* AcR42, plant treated only with phytopathogenic *Cmm* AcR42.

## Data Availability

The data of 16S ribosomal gene sequence of Bacillus cereus Amazcala is located in the following link: https://www.ncbi.nlm.nih.gov/nuccore/1654038701 (accessed on 6 February 2021).

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
