# Peer review of "A Novel Isolate of Bacillus cereus Promotes Growth in Tomato and Inhibits Clavibacter michiganensis Infection under Greenhouse Conditions"

_plants, 2021, doi:10.3390/plants10030506_

Round 1
Reviewer 1 Report
Dear Editors,
The manuscript no. plants-1120675 reports interesting results on the isolation of a bacterial isolate from roots of castor bean (Ricinus communis). The isolate was characterized for plant growth-promoting traits (i.e. phosphate solubilization, gibberellic acid production) and molecularly identified. Subsequently, the isolate was applied in tomato plants as biostimulant and biocontrol agent against Clavibacter michiganensis, under greenhouse conditions.
The purpose of the study is interesting and could be of interest to the readers of Plants. However, the manuscript presents several problems related mainly to the discussion section. The English language should be revised for several grammatical errors (e.g. typos, too long phrases, italics format).
For these reasons, I think that the article should be classified as “major revise”
MAJOR COMMENTS
The introduction contains sufficient details and put the work in the correct contest.
In the Material and Methods section methodologies are well described.
The presentation of the results is clear; however, I would ameliorate the quality of the Figures.
Authors did not discuss the results properly. Authors should discussion better the findings in perspective of the literature and the working hypothesis. Authors should also present the limitation of the phosphate solubilization assay carried out. The use of Pikoskaya medium as a sole source to determine phosphate solubilization is a methodological mistake. In the literature of in vitro screening, it is common to use triphosphate calcium as the substrate, although it has very little to do with reality because strains capable of degrading it are not necessarily PSBs that promote plant growth. To establish the real capacity of phosphorus solubilization by a strain, there is a need to test with a harder-to-dissolve compound, such as Fe-P or Al-P. For a more comprehensive explanation, see Bashan et al 2013 Biol. Fertil. Soils 49:465-479.
Moreover, authors should describe more in deep the implication of the findings obtained and presents future perspectives.
All the manuscript should be revised for the English language. There are many typos and grammatical errors.
MINOR COMMENTS
Revise italics for scientific names, in vitro and in vivo.
Revise superscripts
I hope that my contribution could be useful for your final decision.
Best regards,
Author Response
We want to thank the worthy reviewer´s comments to our paper. We prepared a new manuscript in which we incorporated all the corrections and suggestions requested by the 3 reviewers. The English language was revised by a native-english speaker (Dr. Marieke Vanthoor), all scientific names were rewritten in italics, superscripts were placed where needed and the rest of the suggestions are described below:
Reviewer 1
Comment: Authors should discussion better the findings in perspective of the literature and the working hypothesis.
Address: The discussion section was improved in the revised version.
Comment: Authors should also present the limitation of the phosphate solubilization assay carried out.
Address: According to the references suggested the problem could be a false negative if the halo doesn’t appear, but in our case the halo appeared and in complement with the other tests we could say that the isolated is a PGPB
Comment: Authors should describe more in deep the implication of the findings obtained and presents future perspectives.
Address: This description was improved in the revised version
The rest of the comments are included in the revised manuscript.

Reviewer 2 Report
This manuscript concerns the effects of a new bacteria, Bacillus cereus strain “Amazcala”, on tomato growth and against some bacterial strains of the plant pathogen Clavibacter michiganensis in order to study if the new bacteria could be considered a PGPB. Though the objective is good there are several flaws in the methodology, in the language used and in the exposure clarity. These flaws do not allow to read the paper smoothly and this often compromises the communication of the research. For this, I do not report any comments about Discussion section. As it is, in my opinion, the manuscript cannot be accepted for publishing.
Following I reported some comments, but more are needed
Throughout the manuscript:
In the ‘Introduction’ section a brief mention on plant defence responses to pathogens should be done.
The aims should be explained better
Please, check if the species names are in italics and if the measurement units have superscripts (such as mL-1, 1x104…).
Why in some experiments the authors used Qbacter® as positive control, and in other PROCOBI? Why they didn’t use the same?
Why a chemical control was used only for the greenhouse experiment? It shoul be included also in in vitro experiment.
Statystical analysis: ‘differences between the means were determined by one-way ANOVA’ it is WRONG.
Why in same cases a Tukey test was used and in other case Kruskal Wallis H test?
In the captions, why do the authors report the n of a single experiment and that of the two experiments? This must be clear and explained better in Statistical analysis. For the caption, I suggest to refer to Bruno and Sparapano, 2006 Effects of three esca-associated fungi on Vitis vinifera.. Physiological and molecular plant pathology 69: 182-194.
Abstract
‘preventive disease assay’ and ‘our results B.c-A displayed features’ not correct. Please, rewrite.
Line 17-19: ‘Plant growth promoting bacteria (PGPB) are an excellent alternative for the reduction of chemical fertilizers in addition to offering protection for biotic and abiotic stress.’ change in
‘..could represent an alternative to chemical biofertilizers and pesticides offering protection for both biotic and abiotic stresses.’
Line 19: ‘Ricinus communis’ in Italics
Line 20: ‘Bacillus cereus’ in Italics
Line 22: change ‘provided’ in ‘caused’
Line 23: ‘Interestingly, B.c-A also significantly..’ change in ‘Moreover, B.c-A significantly decreased the severity..’
Line 24: I suggest to delete ‘cv. Ailsa Craig’ from the abstract. ‘Clavibacter michiganensis’ in Italics
Line 25-27: Change ‘Based on our results B.c-A displayed features to be considered as PGPB and might be an alternative to protect tomato production against Cmm infections and enhance tomato production under greenhouse conditions.’ in ‘Based on our results, B.c-A can be considered as PGPB and a useful tool in Cmm disease control on tomato plant under greenhouse conditions.’
‘enhance tomato production’ I suggest to delete it. It is include in PGPB concept
Introduction
Please, explain PGPB and in particular Bacillus in a little more detail. A wide range of bacterial species are used as PGPB, in which Bacillus species are included; some characteristics of Bacillus members such they are rod-shaped, endospore-forming Gram-positive; some examples of effects of Bacillus species on tomato or other plants and against Clavibacter michiganensis or other plants pathogens.
Line 31-34: ‘The rapid growth of the world population, as well as the need to produce food in a more sustainable way without the use of toxic synthetic agrochemicals and the urgent need to reduce the use of fossil sources, has caused in recent years the necessity to generate search for alternatives to these substances [1].’ I suggest to change in
‘The rapid growth of the world population, as well as the need to produce food in a more sustainable way without the use of synthetic agrochemicals, has brought in recent years to search for alternatives substances [1].’
Line 34: ‘One of these alternatives could be the use of beneficial microorganisms like plant growth promoting bacteria (PGPB)..’
Line 39-43: ‘PGPB´s have the ability to directly stimulate the development of plants by means of several mechanisms such as: nitrogen supply through the process of biological fixation of atmospheric nitrogen, production of growth regulating substances, solubilization of minerals, induction of systemic resistance to pathogens and inhibition of the growth of antagonistic organisms [1] [4].’
Line 43: ‘antagonistic microorganisms’ it is not clear. Do the authors mean harmful microorganisms? Please explain it better.
Line 50: tomato is a major.. Tomato is one of the most cultivated vegetable worldwide.. Please insert a reference
Line 52: CROP?? Maybe vegetable o fruit…? Please, insert a reference
Line 56: ‘ssp.’ No in Italics
Line 57: ‘Lanteigne et al. 2012’ must be transformed in number
Please, introduce better the pathogen: it is a gram positive, it’s the only recognized species of the genus Clavibacter.. and some details about the disease caused by Cmm such as symptoms on plant and on fruit. Please, explain better systemic infection. Which are the symptoms in the internal vasculature? And what about localized infection? what does bring to the progression of the disease? I suggest some articles to improve this part:
Chalupowicz Laura, et al. Differential contribution of Clavibacter michiganensis ssp. michiganensis virulence factors to systemic and local infection in tomato. Molecular plant pathology, 2017, 18.3: 336-346.
Nandi, Munmun, et al. "Clavibacter michiganensis ssp. michiganensis: bacterial canker of tomato, molecular interactions and disease management." Molecular plant pathology 19.8 (2018): 2036-2050.
Line 58: ‘This phytopathogen infects host plants through leaves, stems, roots, wounds..’ this is not correct. This pathogen can enter the host plant through wounds and natural openings in leaves, stems and roots and broken stomata (Reference). In addition, Cmm can infects seed externally or it can enter seed coat and endosperm (reference).
Lines 63-65: I would change the phrase in ‘Moreover, there are no tomato cultivars resistant to Cmm disease and breeding genetic progress for resistant tomato lines is modest
Line 62-67: I think that ‘prevention’ is a fundamental concept since Cmm dissemination can occur through both contaminated seeds and both infected transplants, as well as through Cmm‐infested soil, equipment and tools. The authors should include also this concept.
Lines 65-67: ‘Hence, there is a need to generate technology according to the new demands of the market faced by producers, i.e generating products based on PGPB´s isolates [12]’. It is not clear
In Mexico, is Cmm classified as quarantine organism?
Line 68-71:
I would change ‘The present work aimed to evaluate the protecting effect of a novel isolate of Bacillus cereus strain Amazcala (B.c-A) isolated from roots of castor bean (Ricinus communis) in tomato plants against bacterial canker disease caused by Cmm from Central Mexico under greenhouse conditions.’ In:
‘The present work aimed to evaluate the protecting effect of a novel isolate of Bacillus cereus strain Amazcala (B.c-A) isolated from roots of castor bean (Ricinus communis) against bacterial canker disease caused by Cmm on tomato plants under greenhouse conditions.’
Lines 71-75: You suldn’t mention the results here
Section ‘2.1. Isolation of bacteria’: You should insert the % identity with match strings
193: ‘Figure 7. Typical morphology of the tomato plants preventively’ morphology? I would say symptoms..
Line 102: ‘plantssolation’ Please, correct it.
Line 261: ‘-1’ must be in superscript
Line 269: Please, explain better what ‘by serial dilutions’ means
For ‘Isolation of PGPB’ method, please, is there a reference?
Line 272: delete ‘:’
Line 274: ‘abundance of typical colony morphology’ it’s not clear. It must be explained better.
Lines 297-299: Rephrase
Line 298:’.. overnight growth of the isolates on..’ it is not clear. What is ‘Isolates’ referring to?
Lines 299-304: ‘A commercial strain of Bacillus subtilis (PROCOBI, Queretaro, México) was evaluated as positive control. The formation of transparent zones around the bacterial colonies was an indication of the solubilization of the inorganic phosphate contained in the PVK agar. This analysis was carried out in two independent experiments in triplicate and the solubilization index was determined by measuring the diameter of the transparent zones of the solubilization halo generated in the PVK medium using a vernier [25].’
‘..phosphate contained in the PVK agar..’ where? In Petri dishes? How long after the halo appeared? If the experiments were carried out in Petri dishes, where they were incubated?
‘A commercial strain of Bacillus subtilis (PROCOBI, Queretaro, México) was considered as positive control. The formation of transparent zones around the bacterial colonies was measured and a solubilisation index was…’. An ‘index’ is calculated by giving values in a scale. But where is the index? In Fig. 1 reports the diameter halo in mm!
Line 305: ‘Plant growth promotion studies in tomato’ change in ‘Effect on plant growth
Line 306: ‘tomato seedlings’, Which var??
Line 307-310: Please, rephrase in appropriate way
Line 314: Please, check if ‘1.5X108’ is right
Line 315-317: it is not clear
Line 319: treatment with ‘1x104 CFU mL-1 and 1x108 CFU mL-1’ is not clear. Are the treatment subsequent? Or are they different treatment?
Line 320: ‘substrate of the bacterial suspension was inoculated within the substrate’ too much ‘substrate’. Not clear.
Line 321: ‘where the tomato plants were growing’ When? Please, describe better all the steps in a more clear way.
Line 323: ’subsequent inoculations’ how much inoculum? Substrate treatment as irrigation?
Line 326: ‘amount of chlorophyll’ Where was it determined? In dried leaves??
Line 327: I would change ‘In vitro antagonism tests against Clavibacter michiganensis’ in ‘Antagonism tests against Clavibacter michiganensis in vitro’
Line 328: A pathogen is isolated from plant or plant portion or tissue, not from an area.
Line 329: ‘Three different strains of Cmm were isolated in LB plates from tomato production areas affected by these bacteria in Central Mexico (Acr42, 1565 and sp)’. Were isolated and identified? Depeding on what you can distinguish Acr42, 1565 and sp.?
Line 333: ‘grown either on LB or MM’ Why were used both? Why not only just one?
Lines 334-340: Please rephrase better. It is not clear.
Line 338: The right name is Pseudomonas aeruginosa. What is the sense of including human bacteria pathogens?
Line 341: (n=3, total in 3 experiments n=9) Why only here ‘n’ was specified?
Line 342: I would change ‘Antagonism tests against Clavibacter michiganensis under greenhouse conditions’ in ‘Evaluation of antagonistic activity of B.c-A against Clavibacter michiganensis under greenhouse conditions’
Line 362: I would change ‘Stress response-antioxidant enzyme activities measurements’ in ‘Catalase (CAT), phenylalanine ammonia lyase (PAL) and superoxide dismutase (SOD) enzyme activities
Fig.2 and Fig. 5: There’s not statistical analyses
Line 374: ‘Gene expression-associated to plant defense’ This section must be rewritten in a more appropriate way.
Author Response
We want to thank the worthy reviewer´s comments to our paper. We prepared a new manuscript in which we incorporated all the corrections and suggestions requested by the 3 reviewers. The English language was revised by a native-english speaker (Dr. Marieke Vanthoor), all scientific names were rewritten in italics, superscripts were placed where needed and the rest of the suggestions are described below:
Reviewer 2
We appreciate all the suggestions about the wording; all of them were included in the revised manuscript.
Comment: In the ‘Introduction’ section a brief mention on plant defense responses to pathogens should be done.
Address: a description of plant defense responses were added in the introduction section
Comment: The aims should be explained better.
Address: We improved this explanation in the revised version
Comment: Why in some experiments the authors used Qbacter® as positive control, and in other PROCOBI? Why they didn’t use the same?
Address: Qbacter is an organic chemical product to cope Clavibacter infections. And PROCOBI provided a Bacillus subtilis product with known phosphate solubilizing activity. Thus, we cannot use the same; it depends on the test to be carrying out.
Comment: Why a chemical control was used only for the greenhouse experiment? It should be included also in vitro experiment.
Address: The in vitro experiment was carried out to find an isolate that had the same or greater antagonism to an existing product (comparing bacteria vs bacteria), on the other hand, in the in vivo experiment the objective was to test it against some existing products to cope Clavibacter on the Mexican market including one chemical.
Comment: Statystical analysis: ‘differences between the means were determined by one-way ANOVA’ it is WRONG. Why in same cases a Tukey test was used and in other case Kruskal Wallis H test?
Address: Yes, we agree wth this observation. We corrected in the revised ,manuscript the mistake about one-way ANOVA; means differences were evaluated using Tukey test. Kruskal-Wallis H test was used in the study of canker disease severity, because this is an ordinal variable that in our study did not displayed normality in the data distribution using the Anderson-Darling test, thus, this was the reason why we used a non-parametric test only in this case. In the rest of the experiments we analyzed the data using ANOVA and Tukey test if necessary. This is now included in the revised manuscript in the materials section (Statistical analysis).
Comment: Please, in introduction explain PGPB and in particular Bacillus in a little more detail. A wide range of bacterial species are used as PGPB, in which Bacillus species are included; some characteristics of Bacillus members such they are rod-shaped, endospore-forming Gram-positive; some examples of effects of Bacillus species on tomato or other plants and against Clavibacter michiganensis or other plants pathogens.
Address: This information was added in the introduction section in the revised manuscript.
Comment: Line 43: ‘antagonistic microorganisms’ it is not clear. Do the authors mean harmful microorganisms? Please explain it better.
Address: We replace this for the correct concept. Comment: Please, introduce better the pathogen: it is a gram positive, it’s the only recognized species of the genus Clavibacter. and some details about the disease caused by Cmm such as symptoms on plant and on fruit. Please, explain better systemic infection. Which are the symptoms in the internal vasculature? And what about localized infection? what does bring to the progression of the disease? Address: this information was added, and it is found in lines 52-59.
Comment: Please, rephrase in appropriate way the information in line 307-310 Address: We rephrased properly the requested sentences in the revised manuscript.
Comment: Line 338: The right name is Pseudomonas aeruginosa. What is the sense of including human bacteria pathogens?
Address: To assess the specificity of the antagonistic effect observed against Cmm checking possible effects against other bacteria.
Comment: In Fig.2 and Fig. 5: There’s not statistical analyses.
Address: They are mentioned in the figures, as well as in the statistical analysis section in methods.
Comment: Line 374: ‘Gene expression-associated to plant defense’ this section must be rewritten in a more appropriate way.
Address: This section was rewritten in the revised manuscript.
The rest of the comments are corrected and included in the revised manuscript.

Reviewer 3 Report
Plants:
A novel isolate of Bacillus cereus promotes growth in tomato and inhibits Clavibacter michiganensis infection under green-house conditions.
Minor comments:
- Comment 1:
- Line 24 : Clavibacter michiganensis correct it to italics
- Line 45: those colonize…inside the root., correct it
- Line 56 : Cmm, have different abbreviations in the abstract. Which one is correct? if the abstract Cmm full name is correct no need to mention here.
- Line 63: Tomato….modest, Rewrite
- Comment 2: In the introduction section, there should be some information on how various PGPB are contributing to plant growth and Disease/infection control citing recent studies to strengthen the hypothesis
- Comment 3: Introduction the last paragraph is showing the results of the work, author should state the hypothesis by stating the aim of this work
- Comment 4: Bacillus cereus “Amazcala” (B.c-A) correct it to (B.c-A)
- Line 83: LB full form
- Line 92: This…Test, rewrite
- Line 89: what is the meaning of ca.?
- Comment 5: The author should correct the scientific names to italics in several places of manuscript
- Line 102: Plantssolation? Correct and rewrite so that the sentence will have some meaning
- Line 105-107: However..content., correct the sentence
- Line 119: t 1x 104 CFU.mL-1 (C1) or at 1x 108 CFU.mL-1(C2): 104 and 109 CFU? Correct it
- Line 123-124: the sentence is incomplete
- Line 124: ROS, SOD, CAT full forms?
- Life 129: The latter asseveration also suggested that it is likely that these 128 treated plants might display increased tolerance to diseases as that caused by Cmm. .. How? Explain in detail or discuss in in discussion part
- Line 141: chs correct it to italics
- Comment 6: 2.5 gene expression. Rewrite the entire content in a more understandable way
- Comment 7: 2.7 modify the title
- Comment 8: AcR42 full form?
- Comment 9: 225 to 235 rewrite with proper usage of scientific words instead of using casual words, authors repeatedly used the word “this” in the manuscript try to find an alternative for sentence construction ..., Esp. rewrite 227 to 229
- Comment 9: manuscript cited very less recent citations especially, in the introduction and Discussion. The author needs to describe the findings of this study compared with other recent studies in tomato PGPB and Biocontrol or bacteria that contribute to plant health.
I would recommend the publication of this manuscript after addressing minor changes.
Author Response
We want to thank the worthy reviewer´s comments to our paper. We prepared a new manuscript in which we incorporated all the corrections and suggestions requested by the 3 reviewers. The English language was revised by a native-english speaker (Dr. Marieke Vanthoor), all scientific names were rewritten in italics, superscripts were placed where needed and the rest of the suggestions are described below:
Reviewer 3
Comment: In the introduction section, there should be some information on how various PGPB are contributing to plant growth and Disease/infection control citing recent studies to strengthen the hypothesis.
Address: This information was added in the introduction section of the revised manuscript.
Comment: In line 129 the latter asseveration also suggested that it is likely that these 128 treated plants might display increased tolerance to diseases as that caused by Cmm. How? Explain in detail or discuss in in discussion part.
Address: A possible explanation of this asseveration is now included in the revised version of the manuscript.
Comment: AcR42 full form?
Address: It is the name of the Cmm strain in the microorganism´s collection of the IPICYT, who provided us.
Comment: Manuscript cited very less recent citations especially, in the introduction and Discussion. The author needs to describe the findings of this study compared with other recent studies in tomato PGPB and Biocontrol or bacteria that contribute to plant health.
Address: The requested information was updated in the introduction and discussion sections of the revised manuscript.
The rest of the comments are included in the revised manuscript

Round 2
Reviewer 1 Report
The authors answered correctly all my previous comments. This new version of the manuscript could be considered for publication in the present form.
Author Response
Thanks for your comment.
Reviewer 2 Report
Line 18: 'and pesticides to offering protection for biotic and abiotic ' delete 'to'
Fig. 5: There’s not statistical analyses
Author Response
Thank you for your kind comments. Both suggested corrections have been addressed in the minor revision version now submitted.
